

**What is the stability of additional organic carbon stored thanks to alternative**
**cropping systems and organic wastes products application? A multi-methods**
**evaluation**
Tchodjowiè P. I. Kpemoua[1,2,3], Pierre Barré[2], Sabine Houot[1], François Baudin[4], Cédric
Plessis[1], Claire Chenu[1]
[1] UMR Ecosys, Université Paris-Saclay, INRAE, AgroParisTech, Palaiseau, 91120,
France
[2] Laboratoire de Géologie, UMR 8538, Ecole Normale Supérieure, PSL Research University, CNRS,
Paris 75005, France
[3] Agence de la transition écologique, ADEME, 49004 Angers, France
[4] UMR ISTeP 7193, Sorbonne Université, CNRS, France
**Corresponding author:**
E-mail address: claire.chenu@inrae.fr (C. Chenu)
**Abstract**
The implementation of agroecological practices often leads to an additional soil organic carbon storage
in these soils, of which we aimed to assess the biogeochemical stability. To achieve this, we
implemented a multi-method approach using particles size and density fractionation, Rock-Eval®
thermal analyses and long-term incubation (484 days), that we applied to topsoil samples (0-30 cm)
from temperate luvisols that had been subjected, in > 20 years long-term experiments in France, to
conservation agriculture (CA), organic agriculture (ORG) and conventional agriculture (CON-LC) in
La Cage experiment, and to organic wastes products (OWPs) applications in QualiAgro experiment,
including biowaste composts (BIOW), residual municipal solid waste composts (MSW), farmyard
manure (FYM) and conventional agriculture without organic inputs (CON-QA). The incubations
provided information on the additional carbon stability in the short term (i.e., MRT <2 years) and
showed that the additional soil organic C mineralized faster than the baseline C at La Cage but slower
at QualiAgro. In OWPs-treated plots at QualiAgro, 60-66% of the additional carbon was stored as
mineral-associated organic matter (MAOM-C), and 34-40% as particulate organic matter (POM-C). In
CA and ORG systems at La Cage, 77-84% of the additional carbon was stored in MAOM-C, versus 16-
23% as POM-C. Management practices hence influenced the distribution of additional carbon in
physical fractions. Utilizing the PARTYSOC model with Rock-Eval® thermal analysis parameters, we
found that most, if not all, of the additional carbon belonged to the active carbon pool (MRT ~ 30-40
years). In summary, our comprehensive multi-methods evaluation indicates that the additional soil
organic carbon is less stable over decadal and pluri-decadal time-scales compared to soil carbon under
baseline practices. Our results show that particle size and density fractions can be heterogenous in their



biogeochemical stability. On the one hand, while additional carbon was mainly associated with MAOM, we suggest that it has a mean residence time exceeding ~30 years, rather than of ≈ 50 years. On the other hand agroecological practices with equivalent additional carbon stocks (MSW, FYM vs CA) exhibited a higher proportion of additional carbon in POM-C under MSW (40%) and FYM (34%) compared to CA (16%), which suggests a high chemical recalcitrance of POM-C under OWPs management relative to CA. Additional soil organic carbon deriving from organic wastes, i.e., biomass that has been partially decomposed and transformed through its processing prior to its incorporation in soil, would be more biogeochemically stable in soil than that deriving directly from plant biomass. The apparent contradictions observed between method can be explained by the fact that they address different kinetic pools of organic C. Care must be taken to specify which range of residence times is considered when using any method intending to evaluate the biogeochemical stability of soil organic matter, as well as when using the terms stable or labile. In conclusion, the contrasting biogeochemical stabilities observed in the different management options highlight the need to maintain agroecological practices to keep these carbon stocks at a high level over time, given that the additional carbon is stable on a pluri-decadal scale.

*Keywords*: soil organic carbon, additional carbon, agroecological practices, Rock-Eval®, biogeochemical stability, incubation, physical fractions.

## 1   Introduction

Soil organic matter (SOM) plays a crucial role in the functioning of terrestrial ecosystems and can contribute to mitigate climate change. A minor change of soil organic carbon (SOC) content can make a significant difference to global climate because soil contains more carbon (C) than vegetation and atmosphere combined (Lal, 2004). The 4p1000 initiative encourages the implementation of agricultural practices that increase and/or maintain soil carbon stocks (www.4p1000.org, Rumpel et al., 2020). At the field scale, changes in SOC stocks result from an imbalance between C inputs (crop residues, litter, root exudates, exogenous organic matter such as organic wastes products (OWPs) and C outputs from the system due to crop residue export, SOC mineralization, leaching, or erosion (Lal, 2018). Although some agricultural practices can reduce mineralization rates (e.g., reduced tillage, see review by Haddaway et al., 2017), it is generally accepted that the most effective way to increase SOC stocks is to increase C inputs (e.g., Virto et al., 2012; Autret et al., 2016; Fujisaki et al., 2018; Chenu et al. 2019). This can be achieved by increasing biomass production in the field and residue return (e.g., cover crops, Poeplau and Don, 2015; Autret et al., 2016), or by mobilizing external carbon resources such as OWPs (Peltre et al., 2012; Paetsch et al., 2016).



The implementation of selected agroecological practices and systems such as conservation
agriculture, agroforestry, OWPs application allows for additional carbon storage in soils (Peltre et al.,
2012; Autret et al., 2016; Paetsch et al., 2016; Pellerin et al., 2019; Bohoussou et al., 2022). However,
knowledge on the biogeochemical stability of this additional carbon is lacking, questioning the
reversibility of this storage. The carbon sink effect will indeed be more effective if the additional carbon
storage is realized in the form of persistent organic carbon (OC) and not in the form of labile OC. We
propose to evaluate and compare the biogeochemical stability of recently stored organic carbon pools
following implementation of various agroecological practices.

Several methods have been reported in the literature to assess the organic carbon temporal
stability in soils. These methods isolate kinetic pools or carbon fractions with contrasting mean residence
times (MRT), e.g., particle size fractionation, Balesdent, 1996; density fractionation, Sollins et al., 2006;
sequential extraction, Heckman et al., 2018; thermal analysis, Barré et al., 2016 and incubation, Schädel
et al., 2020. Physical fractionation is probably the most used method so far to evaluate SOM stability.
Physical fractionation methods isolate fractions based on size, density, or a mixture of both (Chenu et
al., 2015). In a study comparing several fractionation methods, Poeplau et al. (2018) found that particle
size fractionation was well suited to isolate POM fractions from MAOM with contrasting MRT.
Fractionation of SOM into POM and MAOM components can reveal insights about the sources and
stability of SOC (Kim et al., 2022). However, some studies have shown that SOC fractionation methods
fail to accurately separate stable SOC from active SOC, and in particular that the isolated MAOM
fractions are mixtures of labile SOC (MRT of months to year) and stable centennial SOC (Balesdent et
al. 1987; Jastrow et al., 1996; Sanderman et al., 2013; Torn et al., 2013; Balesdent, 1996; Hsieh, 1992;
von Lützow et al., 2007; Sanderman and Grandy, 2020). This may be due to methodological challenges
as much as the fact that there are multiple pathways for SOM formation and stabilization (Cotrufo et al.,
2013; Sokol et al., 2019).

Thermal analysis techniques, long used in petroleum exploration and clay mineralogy, offer a
promising alternative or complement to physical and chemical fractionation methods, and are
increasingly applied to studies of SOC stability (Peltre et al., 2013; Plante et al., 2009). Indeed, several
parameters obtained using thermal analysis are strongly related to SOM biogeochemical stability (Barré
et al., 2016; Poeplau et al., 2019). However, these parameters do not allow us to separate the kinetic
carbon pools (Schiedung et al., 2017). And so, recently, Cécillon et al. (2018, 2021) developed a
machine-learning model, called $PARTY_{SOC}$, showing that Rock-Eval® parameters can be used to
predict the fraction of SOC that is stable at a centennial timescale. Kanari et al. (2022) evidenced that
SOC fractions calculated using $PARTY_{SOC}$ matched the "stable" and "active" OC pools of the AMG
model, i.e., with an MRT of several centuries and ca. 30 years respectively, a model widely validated to



simulate SOC stock evolution in French and European croplands (Clivot et al., 2019; Bruni et al., 2022).
As a result, one can consider that a Rock-Eval® analysis associated to the PARTY$_{SOC}$ model allows for
the quantification of carbon fractions that are stable at a centennial timescale and "active" *sensu* AMG
model.

The incubation method is, however, the only direct test for the biological stability of SOC, that

results from chemical resistance to decomposition and/or organo-mineral associations and/or
inaccessibility of organic substrates to microbial decomposition. Long-term incubations (months to
years) may diverge from the conditions prevailing in the soil profile but provide insights into the
potential decomposability of slower-cycling SOC (e.g., Schädel et al., 2014). In early laboratory
incubations, fast-cycling C respiration dominates total respired SOC, but rapidly declines, while slow-
cycling SOC accounts for most of the respired SOC after the fast SOC pool is depleted.

These different methods do not separate similar carbon kinetic pools. Indeed, the incubation

method isolates carbon with MRT ranging from days to years while others isolate carbon with longer
MRT (decades to centuries). Thus, a multi-method approach will further improve our knowledge of the
biogeochemical stability of SOC in the short, medium or long term. The objective of this study was to
evaluate the biogeochemical stability of additional SOC stored upon the implementation of C storing
agroecological practices using a multi-method approach. To do so, we characterized SOM using particle
size and density fractionation, Rock-Eval® (RE) thermal analysis and incubation in soil from plots
managed using various agroecological practices such as addition of OWPs (composts and farmyard
manure) and alternative cropping systems including no tillage, permanent cover crop and the
introduction of legumes in the rotation.  The application of OWPs is likely to provide organic matter
(OM) that has been pre-stabilized by the storage (manure) or composting process and is hence less
decomposable than the fresh matter provided by plant biomass in alternative cropping systems. Then,
we hypothesized (i) that the biogeochemical stability of additional SOC depends on the management
practices implemented and (ii) that the additional SOC originating from OWPs would be more stable
than that directly originating from plant biomass, but, (iii) overall, that the additional SOC would be less
stable than the SOC stored in the baseline practices.
## 2    Materials and methods
### 2.1    Field Site and soil sampling

This study focuses on two French long-term experiments (LTEs) developed on Luvisols in the

same region, where agroecological practices including conservation agriculture, organic agriculture and
OWPs application (composts and manure) were implemented.



*La Cage experiment* is conducted in Versailles (48°48'N,2°08'E, alt 120 m). During the studied period (1998-2020), the mean annual temperature, precipitation and potential evapotranspiration were 11.6 °C, 633 and 653 mm respectively. The soil is a well-drained deep Luvisol (IUSS Working Group WRB, 2006). The field experiment is arranged in a randomized complete block design, divided into two blocks, themselves divided into four plots for each cropping system. Each plot is divided into two subplots of 0.56 ha, so that two different crops of the crop rotation are present each year, wheat being grown every year in one of the two subplots (Autret et al., 2020). A detailed presentation of crop rotations, soil management and fertilization were given by Autret et al. (2016). The 4 year's crop rotation mainly consisted of rapeseed (*Brassica napus L.*), winter wheat (*Triticum aestivum L.*), spring pea (*Pisum sativum L.*) and winter wheat.

- CON-LC is characterized by a soil and crop management representative of the Paris Basin cereal production, with annual soil ploughing, the absence of organic amendment, a mineral N fertilization (average rate = 143 kg N ha$^{-1}$ yr$^{-1}$) and a systematic use of pesticides.
- CA includes a permanent soil cover, initially fescue (*Festuca rubra*) and since 2008 alfalfa, grown under the main crops, except pea. In the rotation, rapeseed is replaced by maize (*Zea mays L.*) in CA and direct seeding is performed.
- ORG is characterized by alfalfa-alfalfa-wheat-wheat rotation and no synthetic fertilizers nor pesticides.

*The QualiAgro experiment* is located at Feucherolles, 20 km west of Versailles (48°52'N, 1°57'E, alt 150 m). The soil is a Luvisol (WRB, 2015) and cultivated for 21 years with a conventional wheat-maize rotation (Peltre et al., 2012). The average annual rainfall and temperature for the last 20 years is 614 mm and 11 ° C, respectively. It is a field experiment conducted in collaboration with INRAE and Veolia Environment Research and Innovation since 1998, on which composts of OWPs are applied every 2 years for a dose equivalent to ~4 t C. ha$^{-1}$ from 1998 to 2013 and ~2 t C. ha$^{-1}$ from 2015 to 2020. The unit plots are 10 x 45 m$^2$. Each treatment has 4 replicates and OWP are applied every two years on wheat stubble. Since 2015, wheat and maize residues are buried in the soil. Tree organic treatments are considered in this study and compared to a conventional agriculture system without organic inputs (CON-QA):

– Biowaste compost (BIOW): composting of the fermentable fraction of selectively collected household waste, mixed with green waste;
– Municipal solid waste compost (MSW): composting of the residual fraction of household waste after selective collection of packaging;
– Farmyard manure (FYM) which represents a reference amendment.



At both sites, four replicate plots were available per treatment. From each plot, 3 sub-samples were taken
from the topsoil 30 ± 1 cm (in September 2019 at QualiAgro and in November 2020 at La Cage),
thoroughly mixed and combined into one sample. The samples were sieved to 4 mm, homogenized, the
plant material was removed and the soil was oven dried at 35°C for 72h before particle size and density
fractionation and RE thermal analysis.

**Table 1** Soil organic C (SOC), total N (SON) and C/N measured in topsoil. Values in brackets are
standard deviations. CON-QA: conventional agriculture without organic inputs, BIOW: biowaste
compost, MSW: municipal solid waste compost, FYM: farmyard manure, CON-LC: conventional
agriculture, CA: conservation agriculture and ORG: organic agriculture.

| Site | Soil textured | Agricultural Practices | SOC content g.kg$^{-1}$ | SOC stocks t C. ha$^{-1}$ | SOC gain (%) | SON g.kg$^{-1}$ | C/N |
|---|---|---|---|---|---|---|---|
| **La Cage**  | Luvisol 17% Clay 58% Silt 25% Sand | CON-LC | 9.82 ± 0.48 | 42.22 ± 2.08 | - | 1.01 ± 0.07 | 10.58 ± 1.58 |
| | | ORG | 10.39 ± 0.42 | 44.66 ± 1.80 | 6 | 1.09 ± 0.03 | 9.52 ± 0.12 |
| | | CA | 13.30 ± 1.05 | 57.17 ± 4.53 | 35 | 1.29 ± 0.10 | 10.29 ± 0.28 |
| **QualiAgro** | Luvisol 15% Clay 78% Silt 7% Sand | CON-QA | 9.92 ± 0.63 | 39.31 ± 2.49 | - | 0.97 ± 0.08 | 10.35 ± 1.61 |
| | | MSW | 13.84 ± 0.16 | 54.03 ± 0.59 | 33 | 1.35 ± 0.04 | 10.26 ± 0.42 |
| | | FYM | 13.91 ± 0.37 | 54.77 ± 1.40 | 42 | 1.36 ± 0.02 | 10.21 ± 0.37 |
| | | BIOW | 16.04 ± 0.68 | 63.17 ± 2.56 | 64 | 1.62 ± 0.01 | 9.87 ± 0.41 |


**2.2    Calculation of SOC stocks**
SOC stocks were calculated at equivalent soil mass in both long-term experiments. Thus, at
QualiAgro the SOC stock was calculated by multiplying the SOC content by bulk density (data provided
by QualiAgro) and was normalized to a depth of 10 cm (factor $10^{-3}$) (reference soil mass of 3963 kg. ha$^{-1}$
$^{1}$). Bulk densities between 1998 and 2019 increased significantly in all plots. We calculated the
additional soil thickness required to achieve this equivalent soil mass in treatments with lighter tilled
layers as described by Ellert and Bettany (1995):
$$T_{add} = \frac{\left((M_{soil\ equiv} - M_{soil\ topsoil}) * 10^{-4}\right)}{\rho b_{subsoil}} , (1)$$
where $T_{add}$ is the additional thickness of the sub-soil layer expressed in cm needed to reach the equivalent
soil mass, $M_{soil\ equiv}$ is the equivalent soil mass of the denser horizon in kg. ha$^{-1}$. In our study, the dense



0-29 cm layer was the reference treatment in 2019 with a bulk density of 1.37 g.cm$^{-3}$ giving an equivalent
soil mass ($M_{\text{soil equiv}}$) of 3963 kg. ha$^{-1}$. $M_{\text{soil topsoil}}$ is the soil mass in the surface (tilled) layer and ρb sub-
soil is the bulk density of the underlying 29-35 cm layer (in g.cm$^{-3}$). Carbon stocks per hectare in
equivalent soil masses (Stock $_{\text{C equiv}}$) were calculated by adding the carbon stock in the surface layers
(Stock $_{\text{C topsoil}}$) and in the additional underlying layers (Stock $_{\text{C, Tadd}}$) with the following formula:
$$\text{Stock}_{\text{C equiv}} = \text{Stock}_{\text{C topsoil}} + \text{Stock}_{\text{C Tadd}} \quad , (2)$$
At La Cage experiment, the sample was taken at the depth equivalent to a soil mass of 4300 kg.
ha$^{-1}$. The carbon stocks were calculated by multiplying the SOC contents with this equivalent soil mass.
We then calculated the additional carbon storage (ΔSOC stock) considering each time the CON-LC
baseline at La Cage and the CON-QA baseline at QualiAgro. The following formula was used:
$$\Delta\text{SOC stock} = \text{Stock}_{\text{C Practice}} - \text{Stock}_{\text{C baseline}} , (3)$$
With Stock $_{\text{C Practice}}$: the carbon stock of the agroecological practice and Stock $_{\text{C baseline}}$: the carbon stock
of the baseline. The standard deviation used for the additional carbon stock was calculated based on the
equation described by Kuzyakov and Bol, (2004) as follows:
$$SD_{\Delta\text{SOC}}\text{stock} = \sqrt{(\text{SD}_{\text{stock C Practice}})^2 + (\text{SD}_{\text{stock C baseline}})^2}, (4)$$
### 2.3  Particle size and density fractionation
The method uses a preliminary disaggregation aiming at the best compromise between maximum
destruction of micro-aggregates of size < 50 μm, and respect of the integrity of organic debris (Balesdent
et al., 1991) and combines fractionation by particle size to separate POM from OM associated with clays
and silts minerals with water flotation to separate POM from sands. For this purpose, approximately 50
g of soil was suspended in 180 mL of 0.5% sodium hexametaphosphate (SHMP) saline solution in a 250
mL polyethylene bottle; 10 glass beads were added and the whole set to agitation by inversion (REAX
2 type inversion mixers) for 16 hours, at a speed of approximately 50 rpm to destroy the aggregates. The
SHMP solution and the glass beads allows to completely disperse soil aggregates > 50 μm diameter in
these soils (Balesdent et al., 1991). After agitation, the suspension was first sieved on a 200 μm sieve
from which the refusal, the coarse fraction, was recovered in a 250mL glass beaker. We separated the
coarse POM (cPOM) from the coarse sands (cSand) by flotation in water. The suspension <200 μm in
a second time was submitted to a second sieving at 50μm and the same operations were performed to
separate the fine POM (fPOM) from the fine sands (fSand) using water flotation. The suspension <50
μm is submitted to ultrasounds by imposing an energy of 300 J.mL$^{-1}$ necessary to disperse the micro-
aggregates (Balesdent et al., 1998). After this step, we sieved the suspension <50 μm to 20 μm to recover
the coarse silts of size between 20 and 50μm (cSilt) remaining on the sieve. The suspensions containing
particles <20 μm were pooled in a 2L beaker. The separation of the fine silts between 2 and 20μm (fSilt)
from the clays was performed by centrifugation of the <20 μm filtrate at 64 g (circa 500 rpm) for 10min.
The supernatant containing the clays was collected in a 5L beaker. The same process was repeated 4 to
5 times by resuspending the pellet for an optimal recovery of fine silts by decantation. The supernatant





collected in the 5L beaker constitutes the clay fraction (<2 μm) and the pellet after repeated
centrifugation constitutes the fine silt fraction. To reduce the volume of the clay suspension to be freeze-
dried, we added $CaCl_2$ to flocculate the clay particles and by centrifugation for 20 min at 16000 g (circa
8000 rpm) we recovered the pellet which constitutes the clay fraction. An aliquot of the supernatant was
taken to determine the dissolved organic carbon (DOC).

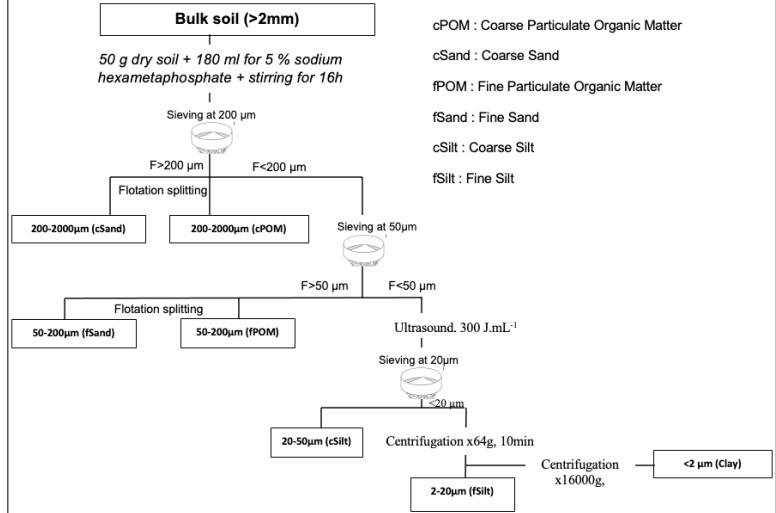


**Fig. 1** Particles size and density fractionation protocol (adapted from Balesdent et al., 1998)
### 2.3.1 Fractions preparation and elemental analysis (C, N)
The fractions obtained during this fractionation were dried or freeze-dried. The organic (cPOM,
fPOM) and mineral (cSand, fSand, cSilt) fractions were oven dried at 50°C for 3 days, while the fine
silt (fSilt) and clay (Clay) fractions were freeze dried. Each fraction was weighed and C, N were
determined using dry combustion (elemental analyzer, Elemantar Vario ISOTOP).
### 2.4 Rock-Eval® (RE) thermal analysis

We analyzed 28 samples of bulk soil, using a RE6 Turbo apparatus (Vinci Technologies). A small
amount of soil (about 60 mg) was required for the analysis, which was performed in two consecutive
steps, during which carbon-containing effluents were directly detected. First, the sample underwent
pyrolysis in an inert atmosphere ($N_2$), followed by oxidation in the presence of $O_2$ (ambient air). The
heating routine applied during pyrolysis was that proposed by Disnar et al. (2003) and Baudin et al.
(2015), including a three-minute isotherm at 200 °C, followed by a 30 °C·min$^{-1}$ heating ramp to 650 °C.
Oxidation began with a one-minute isotherm at 300 °C, followed by a 20 °C·min$^{-1}$ heating ramp to 850
°C and a final five-minute isotherm at 850 °C (oxidation routine presented in Baudin et al. (2015) as the
"bulk rock/basic" method). Simultaneous detection of effluents during both analytical steps generated a



total of five thermograms per sample describing the evolution of hydrocarbons during pyrolysis
(HC_PYR), and CO and $CO_2$ during both pyrolysis and oxidation steps (CO_PYR, $CO_2$_PYR, CO_OX,
$CO_2$_OX).

### 2.4.1    Rock-Eval® parameters

The classical Rock-Eval® parameters were acquired using the RockSix software (Vinci
Technologies) with a good reproducibility (Pacini et al., 2023). They include: six automatically
generated "peaks" defined as specific areas of the three pyrolysis thermograms (S1, S2, S3, S3', S3CO
and S3'CO; Lafargue et al. (2018)), the amount of pyrolyzed carbon (PC corresponding to the sum of
organic C released as HC, CO, and $CO_2$ during pyrolysis), total organic carbon (TOC corresponding to
the amount of organic C released during analysis), inorganic carbon (MinC corresponding to the amount
of C released from carbonate cracking), hydrogen index (HI corresponding to the ratio of hydrocarbons
released to TOC), and oxygen index ($OI_{RE6}$ corresponding to the ratio of organic oxygen released to
TOC). In addition, other parameters used as predictors by the PARTYSOCv2.0EU model were
calculated    based    on    thermograms    obtained    using    R    scripts    available    on    Zenodo
(https://zenodo.org/record/4446138#.YDe84Xlw2SQ) (Cécillon et al., 2021, Kanari et al., 2021). These
include: PseudoS1 (the sum of carbon released during the first 200 s of isothermal 200°C pyrolysis as
HC, CO, and $CO_2$), the S2/PC ratio (the ratio of the amount of hydrocarbons released excluding the first
200 s of pyrolysis to the pyrolyzed carbon), the PC/TOC ratio, the HI/OIRE6 ratio, and ten temperature
parameters (e.g., T30, T50, T70, T90) that describe the evolutionary steps, i.e., at what temperature 30,
50, 70, and 90% of a given gas was released. A detailed description of the definition, units, and equations
used to calculate all parameters can be found in the study of Kanari et al. (2021). The HI and OIRE6 are
commonly reported indices that represent proxies of the SOM H/C and O/C ratios respectively.

### 2.4.2    PARTY$_{SOC}$ model based on Rock-Eval® (RE)

In this study, we used the random forest model based on RE results PARTY$_{SOCv2.0EU}$
(https://zenodo.org/record/4446138#.YDe84Xlw2SQ) proposed by Cécillon et al. (2021). This model
was calibrated on data from 6 long-term agricultural experiments including a bare fallow treatment in
northwestern Europe and can predict the proportion of persistent SOC at a centennial timescale in topsoil
samples (0-30 cm). The model requires a set of 18 RE parameters (e.g., Kanari et al., 2021) characteristic
of a sample and provides a prediction of the proportion of stable SOC for soils from the La Cage and
the QualiAgro long-term experiments. The 18 RE parameters retained were the RE temperature
parameters T70HC_PYR, T90HC_PYR, T30$CO_2$_PYR, T50$CO_2$_PYR, T70$CO_2$_PYR, T90$CO_2$_PYR,



T70CO_OX, T50CO$_2$_OX, T70CO$_2$_OX, and T90CO$_2$_OX and the RE parameters PseudoS1, S2, S2 /
PC, HI, HI / OI$_{RE6}$, PC, PC / TOC$_{RE6}$, and TOC$_{RE6}$ (Cécillon et al., 2021).

## 2.5   Long-term incubation

Polyvinyl chloride (PVC) cylinders 5.7 cm in diameter and 4 cm in height with 2 mm perforations
were used to build soil microcosms. A 50 μm mesh fabric at the bottom of the cylinder supported the
soil while promoting gas exchange. Each cylinder was weighed empty and then with fresh soil
equivalent to 100 g of dry soil. The soil samples were then brought to a bulk density of 1.3 g.cm$^{-3}$ with
a hand press and mold. Knowing the initial water content, the samples were gradually brought to pF 2.5
by adding water with a Pasteur pipette. Then, the microcosms were mounted in 0.5 L jars.  The soil
cylinders were placed on PVC racks and 15 mL of water was added to the bottom of the jars to stabilize
the moisture. The jars were sealed and the whole set was placed in the incubator at 20°C for one week
pre-incubation. Four replicates per agricultural practice were prepared.

After the pre-incubation period, we readjusted the water content of the soil cylinders to pF 2.5
when necessary. A total of 28 soil cylinders were incubated for 484 days under the same temperature
(20°C) and moisture (pF2.5) conditions.
▪ *Mineralization measurement*
Soil organic carbon (SOC) mineralization in samples from both long-term experiments (LTEs)
was measured nondestructively using a micro gas chromatograph (μGC 490; Agilent Technologie;
USA). Measurements were performed 1, 3, 7, 14, 28, 35 days, then one measurement every 2 weeks
until the sixth month and finally, one measurement every month until the end of incubation. The CO$_2$
emitted is measured in parts per million (ppm). It is then converted to μg C-CO$_2$ g$^{-1}$ of dry soil using the
following formula:  $\mu g\, C - CO_2.g^{-1}\, \text{dry soil} = \frac{CO_2(ppm) * M_c * V_b}{V_M * M_{soil}}$ (5), (Védère et al., 2020).
With CO$_2$ (ppm): amount of CO$_2$ emitted measured by gas phase microchromatograph; Mc: molar mass
of carbon in g.mol$^{-1}$; V$_b$: volume of the jar in L; V$_M$: molar volume of the gas in L.mol$^{-1}$ and M$_{soil}$: mass
of the incubated dry soil in g. The absolute amount of carbon mineralized was expressed per unit of
SOC to obtain the specific SOC mineralization in μg C–CO$_2$/100 μg SOC, i.e., % SOC mineralized
(Kpemoua et al., 2023). To calculate the amount of additional carbon mineralized over the 484 days, we
first calculated the difference in absolute carbon mineralization between the agroecological practice and
the baseline practice. We assume that the extra carbon mineralized in the agroecological practice relative
to the baseline practice comes from the additional carbon. Given the amount of additional carbon
(ΔSOC), we then expressed this extra absolute C mineralization in terms of additional carbon (%ΔSOC).

## 2.6   statistical analysis

All data were tested for normality and homogeneity of variance. Log-transformation was
applied, if the transformation improved the normality and variance substantially. A one-way ANOVA
was used to detect significant differences at the 5% threshold in bulk soil carbon stocks, fractions, carbon



pools and amount of carbon mineralized (Cmin). Once a significant difference was detected, Tukey's
multiple comparison test was used to compare carbon stocks, additional carbon stocks, percentage of
total carbon storage and percentage of additional carbon storage in either bulk soil, fractions and carbon
pools according to agricultural practices. All statistical analyses were completed in R (version 4.0.2).

**3      Results**

**3.1      SOC stocks**

The application of organic wastes products (OWPs), increased SOC contents in soils by 64% in

BIOW, 40% in FYM and 39% in MSW compared to the CON-QA; while, at La Cage, the
implementation of ORG and CA increased SOC contents by 6% and 35% respectively, relative to CON-
LC (Table 1).  OWPs significantly increased carbon stocks at QualiAgro. The SOC stocks were in the
order: BIOW > FYM ≥ MSW > CON-QA (Table. 1). At La Cage, SOC stocks were in this order: CA >
ORG ≥ CON-LC (Table 1).

**3.2      SOC distribution in fractions**

The mass proportion, carbon content and % carbon distribution of the physical fractions after

particle size and density fractionation are presented in the supplementary data Table. S1 and Table. S2.
The distribution of SOC stocks over the fractions obtained, expressed in t C. ha$^{-1}$, is given in Figs. 2a
and 2b. Carbon distribution in baseline practices (CON-LC and CON-QA) showed that 19-22% of
carbon was found in POM fractions, versus 78-81% in MAOM fractions (Figs. 4a and 4b). Overall, most
of the organic carbon was located in the clay fraction (64 -72% SOC, see Table. S1 and S2) regardless
of site and the agricultural practice implemented. The carbon distribution in QualiAgro indicated a
significant increase of SOC stocks in the cSand, fPOM, cSilt and Clay fractions after OWPs application
(p<0.05), while no significant difference was observed in the cPOM, fSand and fSilt fractions (p>0.05).
In La Cage, the implementation of conservation agriculture significantly increased SOC stocks as fPOM
and Clay fractions compared to organic and conventional agriculture which remained statistically equal.

We calculated the distribution of additional carbon (ΔSOC) in the different fractions considering

in each case the baseline practice (CON-QA and CON-LC respectively for the QualiAgro and the La
Cage experiments). The additional carbon stock at QualiAgro was 23.86 ± 1.79 t C. ha$^{-1}$ in BIOW
compared to 15.46 ± 1.43 t C. ha$^{-1}$ in FYM and 14.72 ± 1.28 t C. ha$^{-1}$ in MSW (Fig. 2c). At La Cage, the
additional SOC stock was 14.95 ± 2.49 t C. ha$^{-1}$ in CA compared to 2.44 ± 1.38 t C. ha$^{-1}$ in ORG (Fig.
2d). In terms of percentage, we observed in this experiment that, 60-66% of the additional carbon was
localized in mineral-associated organic matter fractions (MAOM-C), which included the cSilt, fSilt, and
Clay fractions, versus 34-40% in particulate organic matter fractions (POM-C), which included the
cPOM, fPOM, cSand and fSand fractions; whereas, in La Cage experiment, conservation agriculture





significantly increased additional carbon in fPOM and Clay fractions (Fig. 2d). In this experiment, 77-
84% of the additional SOC stock was located in the MAOM-C versus 16-23% in the POM-C. However,
the coarse minerals fractions (cSand and fsand) have a negligible proportion of additional carbon
representing 1% in CA and 0% in ORG at La Cage, while this proportion was 2% in FYM, 5% in MSW
and 7% in BIOW at QualiAgro. Furthermore, among practices with equivalent additional carbon stocks
(MSW, FYM, CA) OWPs application resulted in a higher proportion of additional carbon in POM-C
(MSW: 34%; FYM: 40%) compared to CA (16%).

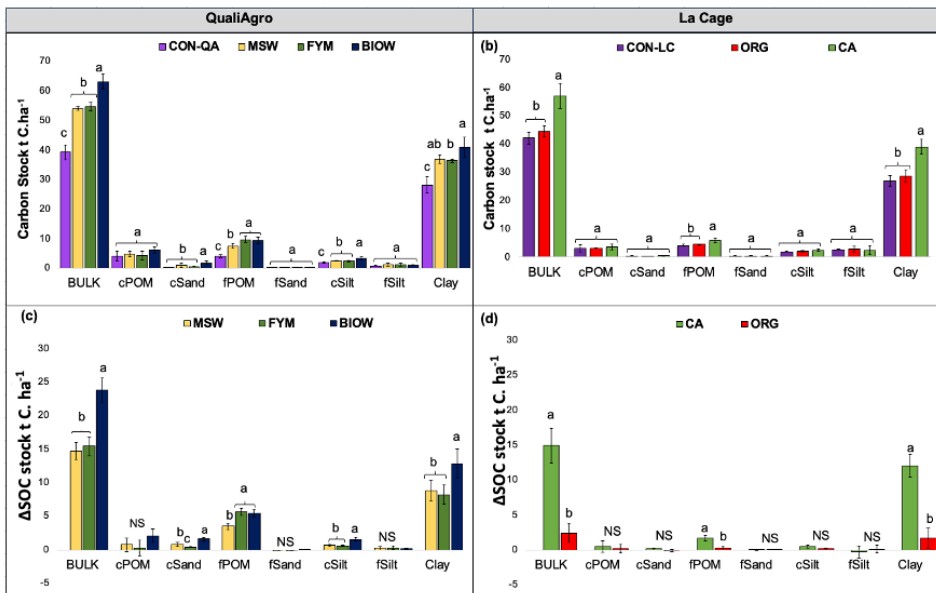

**Fig. 2** Soil organic carbon stock and additional carbon (ΔSOC) stock of bulk soils and physical fractions
(n = 4) at QualiAgro and La Cage experiments. The error bars represent the standard deviations. Grouped
bars with different letters are significantly different between agricultural practices (Tukey-HSD, p <
0.05). CON-QA: conventional agriculture without organic inputs, BIOW: biowaste compost, MSW:
municipal solid waste compost, FYM: farmyard manure, CON-LC: conventional agriculture, CA:
conservation agriculture and ORG: organic agriculture.

### 3.3    Estimating stable and active SOC pools with the PARTY$_{SOC}$ model

The PARTY$_{SOC}$ machine learning model was used to estimate the proportion of stable SOC under
the different managements. The distribution of organic carbon stocks in the active and stable pools are
shown in Fig. 4. In baseline practices, 38-43% of the soil carbon is found in the active pool, versus 57-
62% in the stable pool. The organic wastes products (OWPs) application significantly increased the size
of the active pool (Fig. 3a, ANOVA, p<0.05). It was of 31.87 ± 2.23 t C. ha$^{-1}$ in BIOW compared to
29.62 ± 1.97 t C. ha$^{-1}$ in FYM, 26.97 ± 1.07 t C. ha$^{-1}$ in MSW and 16.76 ± 1.69 t C. ha$^{-1}$ in CON-QA.





The OWPs application significantly increased the size of the stable SOC pool in the BIOW (31.29 ±
0.91 t C. ha$^{-1}$) and MSW (25.15 ± 1.36 t C. ha$^{-1}$) treatments compared to the FYM (25.15 ± 1.44 t C. ha$^{-1}$
$^{1}$) and CON-QA (22.55 ± 1.31 t C. ha$^{-1}$) which were statistically similar. Contrastingly, at La Cage
experiment, 20 years of contrasted management had no significant effect on the size of the stable SOC
pool (28.02 ± 2.95 t C. ha$^{-1}$, 26.31 ± 0.93 t C. ha$^{-1}$, and 26.08 ± 1.89 t C. ha$^{-1}$ for CA, ORG, and CON-
LC respectively). However, CA significantly increased the size of the active pool (29.15 ± 5.79 t C. ha$^{-1}$
$^{1}$) compared to ORG and CON-LC in which it was similar (18.35 ± 3.47 t C. ha$^{-1}$ and 16.14 ± 0.97 t C.
ha$^{-1}$ respectively) (Fig. 3b).


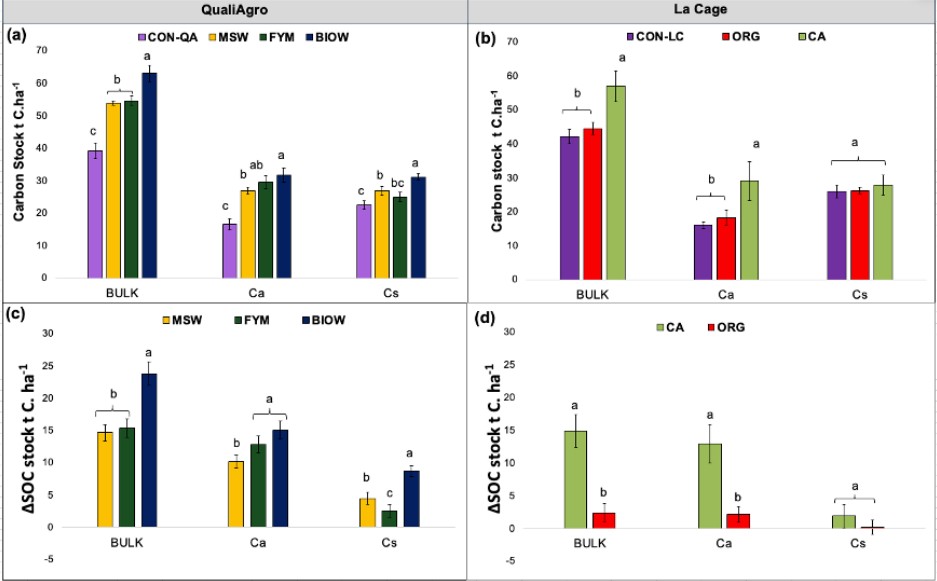


**Fig. 3** Soil organic carbon stock and additional carbon (ΔSOC) stock of bulk soils, active carbon (Ca)
and stable carbon (Cs) (n = 4) at QualiAgro and La Cage experiments. The error bars represent the
standard deviations. Grouped bars with different letters are significantly different between agricultural
practices (Tukey-HSD, p < 0.05). CON-QA: conventional agriculture without organic inputs, BIOW:
biowaste compost, MSW: municipal solid waste compost, FYM: farmyard manure, CON-LC:
conventional agriculture, CA: conservation agriculture and ORG: organic agriculture.

The results of additional carbon distribution in the active and stable carbon pools are shown in

Figs. 3c and 3d. In the QualiAgro experiment, BIOW (15.85 ± 1.48 t C. ha$^{-1}$) and FYM (13.36 ± 1.16 t
C. ha$^{-1}$) had similar active carbon pool size, higher than in the MSW (12.34 ± 0.75 t C. ha$^{-1}$). This active
pool, represented 63-83 % of the additional carbon storage (Fig. 4c). Additional stable carbon pools
ordered as follows: BIOW (8.74 ± 0.79 t C. ha$^{-1}$) > MSW (4.51 ± 0.94 t C. ha$^{-1}$) > FYM (2.60 ± 0.97 t



C. ha$^{-1}$) and represented between 17% (FYM) to 37% (BIOW) of the additional carbon. At La Cage,
87% (CA) to 91% (ORG) of the additional carbon was in the active pool versus 9% (ORG) to 13% (CA)
in the stable pool (Fig. 4d).

### 3.4    Carbon mineralization kinetics
At the end of soil incubation (day 484), the cumulative amounts of mineralized C expressed as
percent soil organic carbon (%SOC) at La Cage experiment, differed significantly between the 3
cropping systems, i.e., 12.60 ± 0.29 %SOC in ORG versus 11.52 ± 1.19 %SOC in CA and 10.21 ± 1.36
%SOC in CON-LC (Fig. 4a). In the QualiAgro experiment, the specific carbon mineralization kinetics
were significantly higher on the baseline practice (CON-QA) without organic inputs compared to the
soils receiving OWPs, where the mineralization kinetics of the MSW and FYM plots were statistically
identical but higher than the BIOW plot (Fig. 4b).

Overall, these two experiments show opposite trends. On the one hand, higher carbon
mineralization under agroecological practices (ORG and CA) in La Cage experiment relative to the
baseline practice (CON-LC), and on the other hand, lower carbon mineralization under agroecological
practices (MSW, FYM and BIOW) in QualiAgro experiment relative to the baseline practice (CON-
QA). Moreover, the percentage of additional carbon mineralized (%ΔSOC) at La Cage in CA (15%) and
ORG (57%) was higher than at QualiAgro (4-5%) (Figs. 4c and 4d). It must be noted however that the
ΔSOC stocks was very small in the ORG treatment, which numerically explains the high %ΔSOC
calculated value.

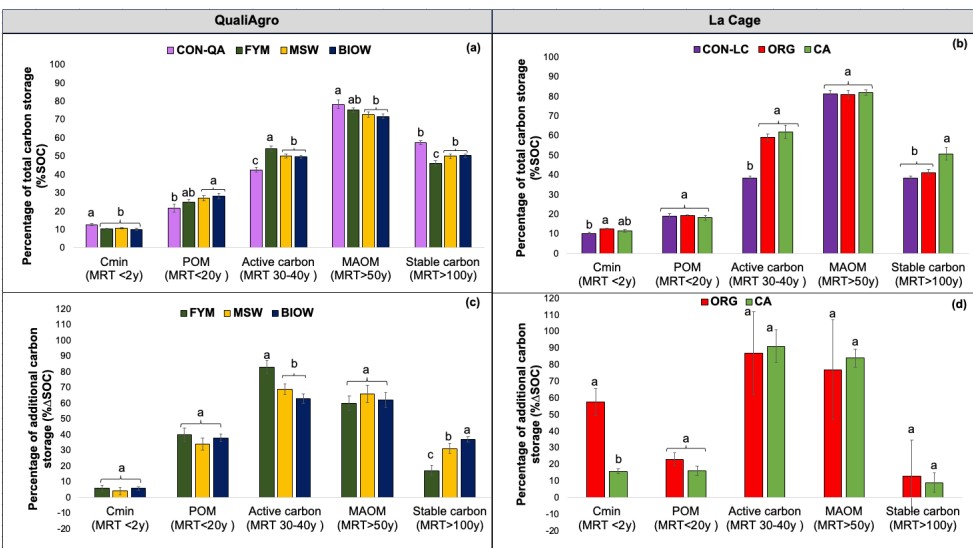

**Fig. 4.** Distribution of total carbon and additional carbon in carbon kinetic pools [Cmin (carbon
mineralized), Active and Stable carbon] or fractions [POM and MAOM] under agricultural practices.



The error bars represent the standard errors. Grouped bars with different letters are significantly different
between agricultural practices (Tukey-HSD, p < 0.05).
## 4   Discussion
### 4.1   Both POM-C and MAOM-C are sensitive to management
The observed distribution of SOC in the fractions, i.e., most of the SOC (70-80%) being located
in the fine fraction (<50 μm), regardless of the agricultural practice implemented (Figs. 4a and 4b), is in
agreement with the literature (Christensen, 1985;1987; Balesdent, 1996; Baldock and Skjemstad, 2000;
Jolivet et al., 2003; Carter et al., 2003; Beare et al., 2014, Curtin et al., 2016, Poeplau et al., 2018).

Many studies have indicated that the POM-C fraction is more sensitive to land use and
management changes than the MAOM-C fraction (Cambardella and Elliot, 1992, Elliot et al., 1994,
Bayer et al., 2001, Salvo et al., 2010). However, our study showed that both POM-C and MAOM-C
fractions were both highly sensitive to the implementation of agroecological practices ca. 20 years. The
application of OWPs resulted in additional soil organic carbon both as POM-C (34-40% of additional
SOC) and MAOM-C (60-66% of additional SOC), while conservation and organic agriculture resulted
in additional organic carbon mainly as MAOM-C (77-84% of additional SOC) and less as POM-C (16-
23% of additional SOC). A significant proportion of the additional carbon is associated with soil
minerals, particularly in the clay fraction (See Table. S1 and Table. S2).

Studies comparing no-tillage versus conventional tillage management showed in the surface layer
an increase in the POM-C fraction with no-tillage and no difference in MAOM-C (Wander et al., 1998,
Hussain et al., 1999; Carbonell-Bojollo et al., 2015; Samson et al., 2020). We therefore ascribe the
observed increase of MAOM-C at La Cage to the introduction of cover crops and the diversification of
species (e.g., legumes) in the rotation. Interestingly, an earlier analysis of SOC distribution at La Cage,
after 5 years of differentiation, showed a significant increase of POM-C in the conservation agriculture
system, while no change of POM-C in the organic system and no significant change of the MAOM-C
(Balabane et al., 2005), suggesting either that 12 years were necessary for the additional POM-C to be
broken down and biodegraded as MAOM-C, or that the introduction of alfalfa as the cover crop instead
of fescue since 2008 (i.e., 12 years later) resulted in more direct rhizodeposits inputs to MAOM-C.
Indeed, according to Autret et al. (2016), the estimated inputs from fescue were lower (0.88 t C. ha$^{-1}$. yr$^{-1}$
) than those coming from alfalfa as a cover crop (1.12 t C. ha$^{-1}$. yr$^{-1}$), about half of these amounts
deriving from root material. The cover crops and legume rotation in conservation agriculture and the
legume rotations in organic agriculture would likely have affected carbon input via the root system as
dead roots (POM) and rhizodeposits (fine sized OM). This would explain the high proportion of carbon
associated with MAOM-C. Typically, the cover crops characterized by low litter quality (e.g., grass)
resulted in higher accumulation of POM that was abundant in plant C, while cover crops with high litter





quality (e.g., legumes) contributed to higher accumulation of SOC (Cotrufo et al., 2013), and of
microbial necromass C (Zhang et al., 2022) in MAOM-C. Thus, the high proportion of carbon in the
MAOM-C at La Cage (77-84% of ΔSOC) compared to QualiAgro (60-66% of ΔSOC) could be
explained by the type and quality of the carbon input. Because, the cover crops increase the time period
in which plant roots interact with the soil environment (Tiemann et al., 2015), they deliver an additional
source of root litter and exudates, providing greater diversity in belowground inputs (Austin et al., 2017).
This promotes the microbial growth and turnover in rhizosphere hotspots, processes that can enhance
the formation of MAOM (Kallenbach et al., 2016).
The recent meta-analysis, conducted by Zhang et al. (2022), indicated that the application of OWPs
significantly increases both MAOM and POM fractions in the soil relative to the control treatment,
which is consistent with our results. Peltre (2010) observed that the short-term application (4 times) of
the OWPs at QualiAgro increased the additional carbon only in the POM-C fraction, the MAOM-C
fraction <50μm being unchanged. Paesch et al. (2016) later found that 7 successive applications of the
OWPs led to additional carbon in the small occluded POM (oPOM$_{small}$) and in the clay fraction. After
11 applications of OWPs we observed an increase in both small POM-C and MAOM-C. This series of
results indicate that the application of the OWPs increase in the short term the POM-C fraction and that
in a longer term (> 10 years) the organic carbon in the POM-C is transferred to the MAOM-C through
biological activity in the soil. The transfer of additional carbon from POM to MAOM is however slower
at QualiAgro compared to La Cage.

### 4.2 POM heterogeneity can mess up SOC stability assessments
The POM in this study consists of crop residues and/or added manure or composts and microbial
residues. The agroecological practices with equivalent additional carbon stocks (MSW, FYM, CA)
showed after 20 years a higher proportion of additional carbon in POM-C under MSW (40%) and FYM
(34%) compared to CA (16%). These results show that it is likely that different management (e.g., OWPs
application, no-tillage, cover crops and legume) alter the way gross organic carbon inputs were
distributed among the different organic carbon fractions. These results can be explained by the fact that
the decomposition rate of organic amendments and the SOC formed and remaining in the long term vary
according to the intrinsic quality of the amendment (Lashermes et al., 2009). For example, Paustian et
al. (1992) observed that high lignin content of FYM, which was more recalcitrant to decomposition,
resulted in greater accumulation of C than lower lignin amendments, such as straw. Previous studies
demonstrated that the OWPs generally are partially stabilized by the composting and storage processes
(Benbi and Khosa, 2014), unlike plant biomass, which is fresh OM.



The incubations revealed that in La Cage experiment, a higher percentage of the additional carbon
was mineralized in conservation agriculture (15% of ΔSOC) over 484 days than additional carbon at
QualiAgro (4-5% of ΔSOC) (Figs. 4c and 4d). The low additional carbon mineralization at QualiAgro
raises questions about the degradability of POM derived from OWPs, which were in higher proportion
(34-40% of ΔSOC) than at La Cage experiment (16-23% of ΔSOC). It is therefore likely that the OWPs-
derived POM were more recalcitrant with higher mean residence times compared to plant-derived POM.
The mean residence time of < 20 years given to POM in the study by Balesdent, (1996) may not be
applicable to systems where pre-processed exogenous OM are applied, because in the formed is based
on situations where organic input were crop residues. Thus, we assume a greater chemical recalcitrance
of POM-C in plots receiving OWPs, thereby reducing decomposers activity and carbon transfer to the
fine soil fraction (<50 μm).

### 4.3 Different methods provide a contrasted evaluation of biogeochemical stability

We used different methods to assess the biogeochemical stability of the additional C stored in soil
thanks to specific management options. The incubation method isolates carbon with MRT ranging from
days to years (MRT < 2y in this case), while particle size and density fractionation isolate carbon
fractions ranging from years to decades (POM with MRT < 20y and MAOM with MRT >50y) and
PARTY$_{SOC}$ model based to RE thermal analysis that isolate carbon pools ranging from decades to
centuries (Active pool with MRT 30-40y and Stable pool with MRT > 100y).

In the QualiAgro experiment, the incubations results indicate greater stability of additional carbon
compared to bulk SOC in the reference system (i.e., lower specific carbon mineralization for soils
receiving OWPs relative to CON-QA). However, the results of particle size and density fractionation
and PARTY$_{SOC}$ based to RE thermal analysis indicate that the additional carbon stored by OWPs
application is on average less stable than the soil carbon in the baseline practice (CON-QA). This is
because, in these plots the additional carbon has a higher proportion of POM (MRT < 20y) and Active
carbon (MRT ~ 30-40y) than the baseline practice (Figs. 4a and 4c). As the incubations target carbon
with MRT of the order of incubation length (i.e., MRT <2y in this study), we posit that this difference
is due to the fact that the different methods do not target the carbon pools with the same MRT. Put
together, these results suggest that, on the scale of a few decades, soil additional carbon in QualiAgro
experiment is less stable than soil carbon in baseline practice, but in a shorter term (i.e., MRT < 2y), the
additional carbon is quite resistant.

In La Cage experiment, the results of the incubations and the PARTY$_{SOC}$ model based to RE
thermal analysis are consistent and indicate that the additional carbon stored by conservation agriculture



and organic agriculture is less stable than the soil carbon in the baseline practice (CON-LC), whereas
the particle size and density fractionation indicates a more stable additional carbon, i.e., a higher
proportion of MAOM (MRT > 50 years) than the baseline practice (Fig. 4). However, the study by von
Lützow et al. (2006) showed that MAOM does not have a unique mean residence time. For example,
land-use change (native and cropped lands) studies have indicated a decrease in carbon content in
MAOM-C over time (Balesdent et al., 1998, Yeasmin et al., 2019). Lutfalla et al. (2019), using samples
from 42 plots in Versailles, observed a decrease in carbon content in the clay fraction (< 2 μm) after 52
years of bare fallow conditions, thus questioning the long-term persistence of carbon associated with
clays and MAOM-C. Our results provide evidence that at least part of the carbon contained in MAOM
may not persist in soils over the long term as shown by others previously (e.g., Balesdent, 1987,
Keiluweit et al., 2015, Lutfalla et al., 2019, Chassé et al., 2021). We therefore hypothesize that the
additional carbon stored in the form of MAOM has a lower MRT than the MAOM in baseline practice.

Based on these results, our hypothesis that the biogeochemical stability of additional carbon is less
stable than the carbon in the baseline practice is not always verified. However, considering that MAOM
is kinetically heterogeneous, then the results of these methods can be reconciled. So, the additional
carbon is overall less stable at a decadal or pluri-decadal timescale than the carbon stored in the baseline
practice in both long-term experiments. Furthermore, taking all these elements and the complementary
nature of the methods into consideration, it emerges that the additional carbon stored thanks to OWPs
application is more stable in the short (MRT < 2y) and long term (MRT > 100y) than the additional
carbon enabled by alternative cropping systems, but less in the decadal and pluridecadal time scale. The
large time scale of SOM persistence shows that qualifying SOC simply as stable or labile is not
sufficient. It is essential to always associate a temporality with the biogeochemical stability that is
described in order to better assess the persistence of carbon in soils.

## 5    Conclusion
This study provided detailed information on the biogeochemical stability of additional carbon
via a multi-methods evaluation. Soils from the same experimental sites but under widely contrasting
management have resulted in contrasting carbon contents and stocks ca. 20 years of management. The
results of particle size and density fractionation and PARTY$_{SOC}$ model suggests that the additional
carbon contained in MAOM may not persist in soils over the long term (> 50 years). Incubation, on the
other hand, provided information on the short-term stability of additional carbon (i.e., MRT <2 years).
Overall, the multi-methods evaluation showed that additional carbon was less stable at the decadal and
pluri-decadal time-scales than carbon under baseline practices. However, incubations and PARTY$_{SOC}$
model based to RE thermal analysis revealed that additional SOC in the QualiAgro experiment was
more stable in short- term (MRT < 2y) and long- term (MRT >100y) than that in La Cage experiment.



Additional SOC deriving from organic wastes, i.e., biomass that has been partially decomposed and
transformed through its processing (digestion by cattle, storage and composting) prior to its
incorporation in soil, would have a different biogeochemical stability than that deriving directly from
plant biomass. Widely used (incubation, particle size fractionation) and increasingly used methods (RE)
provide seemingly inconsistent assessments of the biogeochemical stability of SOC. These apparent
contradictions can be explained by the fact that they address different kinetic pools of organic C. Care
must be taken to specify which range of residence times are considered when using any method
intending to evaluate the biogeochemical stability of SOM, as well as when using the terms stable or
labile. As we found that the additional SOC stored thanks to the implementation of different
management options had contrasted biogeochemical stabilities, there is a need to evaluate the
biogeochemical stability of the additional SOC stored via other management options (e.g., agroforestry,
lengthening temporary leys, no tillage…).
**Competing interests**
The contact author has declared that none of the authors has any competing interests
**Acknowledgements**

This research has been supported by the French Agence Nationale de la Recherche (StoreSoilC project,
grant ANR- 17-CE32-0005). C. Chenu also thanks the CLand programme (ANR-16-CONV-0003). T.
P. I. Kpemoua acknowledges the support of ADEME. The QualiAgro field experiment forms part of the
SOERE-PRO (network of long-term experiments dedicated to the study of impacts of organic waste
product recycling) integrated as a service of the ''Investment for future'' infrastructure AnaEE-France,
overseen by the French National Research Agency (ANR-11-INBS-0001)'. The QualiAgro site is
conducted within a collaboration between INRAE and Veolia. The authors also thank the personnel
from Unité expérimentale Versailles Grignon INRAE, Michel Bertrand for the maintenance of and
access to the La Cage long term experiment and Fabien Ferchaud for the collaboration upon La Cage
SOC stocks assessment.

**Author contribution**

TPIK, CC, PB and SH designed the study. TPIK performed soil fractionation and long-term incubation.
FB and CP performed the RE6 thermal analyses and elementary analyses respectively. PB performed
the R codes to PARTY$_{SOC}$ machine learning model. TPIK wrote the R codes and performed all
statistical analyses. TPIK, CC, PB, SH and FB contributed to the interpretation of the results. TPIK
prepared the paper with contributions from all coauthors.



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
