# Peer review of "What is the stability of additional organic carbon stored thanks to alternative"

_EGUsphere, 2023_

## Author Response (AR1)

Answer reviewer #1

*I recommend accepting this manuscript with minor revisions. The paper provides an insightful investigation into the biogeochemical stability of additional soil organic carbon (SOC) as influenced by various agroecological practices. The multi-method approach adopted, encompassing particle size and density fractionation, Rock-Eval® thermal analysis, and long-term incubation, is commendable for its thoroughness, contributing to a nuanced understanding of SOC dynamics. The incorporation of long-term data (over 20 years) from two French experiments enhances the study's relevance, particularly in understanding the long-term impacts of agricultural practices on SOC stability. The findings are significant for their contribution to soil science and the broader context of sustainable agriculture and climate change mitigation.*

*However, there are areas that require minor revisions for clarity, consistency, and depth of understanding.*

We thank the reviewer for the positive comments. We addressed the issues raised by modifying some parts of the manuscript. Please find the details below.

*Specific Comments for Improvement:*

*Introduction:*

*L38: The sentence is unclear and needs rephrasing for better comprehension.*

Answer L38: We have revised this sentence as you can see in :

L40-L42 in the new version: « On the other hand, although the additional carbon is mainly associated with MAOM, the high proportion of this carbon in the active pool suggests that it has a mean residence time not exceeding ~50 years. »

*L103: The term 'kinetic carbon pools' should be clearly defined.*

Answer L103: We decided to maintain the term "kinetic carbon pool" as it is an appropriate one. We define it as being a subset of the carbon pool to which we associate a residence time and where we consider that this subset has the same mean residence time.

*L105-107: The sentence construction is clunky; consider rephrasing for smoother readability.*

Answer L105-107: We have revised this sentence as you can see in :

L112-L115 in the new version: "Kanari et al. (2022) evidenced that SOC fractions calculated using PARTYSOC matched the stable (MRT >100 years) and active (MRT ~ 40-30 years) OC pools of the AMG model, a model widely validated to simulate SOC stock evolution in French and European croplands (Clivot et al., 2019; Bruni et al., 2022)."

*L110: Clarify the meaning of 'active' when used in quotes.*

Answer L110 : Thank you for your suggestion. In the paragraph above, we specified the residence time of the active pool in the AMG model. We don't think it's necessary to specify this again.

*L122: It would improve readability to explicitly list the 'others' referred to.*

Answer L122: Thank you for your suggestion. We have added some references in the new version. You can see them in the:

L127-L129: "These different methods do not separate similar carbon kinetic pools. Indeed, the incubation method isolates carbon with MRT ranging from days to years (Schädel et al., 2014) while others isolate carbon with longer MRT (decades to centuries) (Cécillon et al., 2018, 2021, Balesdent, 1996)"

*L124: Define what constitutes short, medium, and long term in years (e.g., short (<2 years), medium (2-50 years), long term (>50 years)).*

Answer L124: Thank you for your suggestion. We have taken this into account in the new version of the MS.

See L129-131: "Thus, a multi-method approach will further improve our knowledge of the biogeochemical stability of SOC in the short (<2 years), medium (2-50 years) or long term (>50 years)."

*Methods:*

*Clearly state the duration of the study in years for better comprehension.*

*L152-159: Full names of treatments should be consistently written, as done for the QualiAgro experiment.*

AnswerL152-L159: Full names have been written for La cage experiment practices.

See L158-166: "

- Conventional agriculture (CON-LC) is characterized by a soil and crop management representative of the Paris Basin cereal production, with annual soil ploughing, the absence of organic amendment, a mineral N fertilization (average rate = 143 kg N ha$^{-1}$ yr$^{-1}$) and a systematic use of pesticides.
- Conservation agriculture (CA) includes a permanent soil cover, initially fescue (*Festuca rubra*) and since 2008 alfalfa, grown under the main crops, except pea. In the rotation, rapeseed is replaced by maize (*Zea mays L.*) in CA and direct seeding is performed.
- Organic agriculture (ORG) is characterized by alfalfa-alfalfa-wheat-wheat rotation with annual soil ploughing and no synthetic fertilizers nor pesticides."

*L169: Description of the La Cage Experiment should be consistent, including the control treatment and use of dashes.*

Answer L169: We take account your suggestion in the new version of MS.

*L181 Table 1: Order the treatments in the table description as they appear in the table for clarity.*

Answer L181: We thank you for your suggestion. We incorporate it in the new MS version.

See L190-L193: "**Table 1** Soil organic C (SOC), total N (SON) and C/N measured in topsoil. Values in brackets are standard deviations. CON-QA: conventional agriculture without organic inputs, MSW: municipal solid waste compost, FYM: farmyard manure, BIOW: biowaste compost, CON-LC: conventional agriculture, ORG: organic agriculture and CA: conservation agriculture."

*L187: I always appreciate Stocks calculated using equivalent soil mass. Nice work!*

Answer L187: Thank you for your comment!

*L204-206: The term 'baseline' might be confused with baseline measurements at the start of the experiment. Consider using a term like 'conventional control' or similar.*

Answer L204-206: Thank you for your suggestion. We have replaced "baseline" with "conventional control" in this MS version.

*L236: In the results, where fractions are lumped into POM and MAOM categories, it would be helpful to highlight which fractions classify as POM and MAOM in Figures 1 and 2.*

Answer L236: We added this sentence: "The POM fraction is the sum of the cPOM and fPOM fractions, while the MAOM fraction is the sum of the cSilt, fSilt and Clay fractions" in the titles of Figure 1 and 2 to clarify which fractions were classified as POM or MAOM.

*L242: Provide a rationale for why fSilt and Clay are freeze-dried instead of oven-dried like the other fractions.*

Answer L242: Thank you for your inquiry. The fSilt and Clay fractions were freeze-dried because the quantity of suspensions obtained at the end of fractionation was large and therefore oven-drying could take some time which could affect the carbon in these fractions.

*L297: Consider rephrasing 'stabilize the moisture' to 'maintain humidity'.*

Answer L297: We have taken this into account.

See L312-314 in the new MS version: "Then, the microcosms were mounted in 0.5 L jars. The soil cylinders were placed on PVC racks and 15 mL of water was added to the bottom of the jars to stabilize the humidity."

*L318: This is a very bold assumption. Could the additional carbon (especially nitrogen rich additional carbon) be leading to the priming of older, 'baseline' carbon? You might be correct, but this could use further justification.*

Answer L318: Thank you for this point of attention. Indeed, it's a bold assumption. If we take into account the analogy made to calculate additional carbon with the other methods, it's still comfortable. We are aware that in this mineralized carbon pool we have new carbon and old carbon induced by the priming effect. At the same time, for the other methods, additional carbon is a mixture of new and old carbon.

*Somewhere in methods, SI, or Results it would be good to see % recovery of fraction and thermal data*

Answer: Thank you for your suggestion. In tables S1 and S2 we indicated the % recovery in mass and carbon. Thermal analysis data will be made available on request.

*Results:*

*If OWP is mentioned again, it might be helpful to remind readers of what each treatment stands for.*

Answer: Thank you for your suggestion. We have reminded readers again what the acronyms for each practice stand for in lines L349 to L354 in the new version of the MS.

L349-354: "The application of organic wastes products (OWPs), increased soil organic carbon (SOC) contents in soils by 64% in biowaste composts treatment (BIOW), 40% in farmyard manure treatment (FYM) and 39% in residual solid waste compost treatment (MSW) compared to the conventional control (CON-QA); while, at La Cage, the implementation of organic agriculture (ORG) and conservation agriculture (CA) increased SOC contents by 6% and 35% respectively, relative to conventional control (CON-LC) (Table 1)."

*Clarify whether the additional carbon observed is due to increased stocks or simply reduced losses compared to baseline data.*

We have indicated in the discussion that this additional carbon storage was associated with an increase in carbon inputs through the agroecological practices considered in this study.

*L364-366: Make it clear that comparisons are being made across sites.*

*Figures 2 & 3: Ensure consistency in the order of ORG and CA in legends and bar plots.*

Anwer L364-366: We take this point account! See L377-388 in the new version of MS.

L377-388: "In terms of percentage, we observed that the coarse mineral fractions (cSand and fsand) have a negligible proportion of additional carbon at La Cage, representing 1% in CA and 0% in ORG, while this proportion was raised to 2% in FYM, 5% in MSW and 7% in BIOW at QualiAgro. This non-negligible proportion of carbon in the sand at QualiAgro suggests that not all particulate organic matter (POM-C) has probably been isolated from the sand. Hence, in the following, to define the POM-C fraction class, we combine the fractions of cPOM, fPOM, cSand and fSand together. Thus, we observed in the QualiAgro experiment that, 60-66% of the additional carbon was localized in mineral-associated organic matter fractions (MAOM-C), which included the cSilt, fSilt, and Clay fractions, versus 34-40% in POM-C; whereas, in La Cage experiment, 77-84% of the additional SOC stock was located in the MAOM-C versus 16-23% in the POM-C. Furthermore, among practices with equivalent additional carbon stocks (MSW, FYM, CA), OWPs application resulted in a higher proportion of additional carbon in POM-C (MSW: 34%; FYM: 40%) compared to CA (16%)."

And we corrected the Fig 2 &3 to ensure consistency in the order of ORG and CA in legends and bar plots

*L378: Introduce Fig 4 after Fig 3 for sequential flow.*

Answer L378: We corrected it.

*L401: Clearly indicate that the changes in SOC are relative to control treatments.*

Answer L401: We clarified it.

See L425-429 "The results of additional carbon (i.e., the difference between the active or stable carbon pool of agroecological practices and the active or stable carbon pool of conventional control) distribution in the active and stable carbon pools are shown in Figs. 3c and 3d. In the QualiAgro experiment, BIOW (15.85 ± 1.48 t C. ha$^{-1}$) and FYM (13.36 ± 1.16 t C. ha$^{-1}$) had similar active carbon pool size, higher than in the MSW (12.34 ± 0.75 t C. ha$^{-1}$)."

*L418: The section effectively consolidates information from other sections, enhancing the coherence of the manuscript. It may be beneficial to delineate this section with a dedicated subheading for improved navigability and emphasis.*

Answer L418: Thank you for your suggestion concerning this paragraph. We have put it in this section because we believe that these results concern incubation and so it is a sub-section.

*Fig. 4: The mean residence time (MRT) ranges depicted by the various methods in Figure 4 are insightful. However, it is crucial to clarify the overlaps between these SOC pools, as they are not entirely discrete entities. The current presentation suggests that the cumulative percentage of additional carbon surpasses 100% when these pools are combined. A graphical representation or an extension of the MRT ranges in the figure could better illustrate this overlap. For instance, showcasing the MRT as follows could be more illustrative: Carbon mineralization (Cmin) < 2 years, Particulate Organic Matter (POM) < 20 years, Active Carbon 20-50 years, Mineral-Associated Organic Matter (MAOM) > 40 years, and Stable Carbon > 100 years. Such a representation would effectively convey the interconnections and shared boundaries of these SOC pools.*

Concerning Fig. 4, we have modified it so that the calculation is no longer made for 100% for all the pools. We think that the segmentation you have proposed is a good idea, but in our case, we wanted to show how biogeochemical stability could be considered by different methods. We are aware that the pools may overlap, but we opted to show the results by method.

*Discussion:*

*L449: A recent meta analysis by Prairie et al. 2023 (PNAS) shows that no-till does increase MAOM-C in topsoil*

Answer L449: Thank you for this article suggestion. We found the results very interesting and consistent with our findings. We have therefore integrated it into the revised MS.

See Lines L477-480: "A recent meta-analysis by Prairie et al. (2023) indicated that no-tillage increased both POM-C and MAOM-C fractions in soils when this practice was maintained up to 6 years. However, the increase in the MAOM-C fraction was less important than that of the POM-C fraction"

*L483: Consider alternative explanations, such as those suggested by Cotrufo et al. 2015 (Nat Geo Sci), for the formation of POM and MAOM. Cotrufo et al 2015 shows that POM and MAOM likely form under different biochemical and physical pathways. While there is certainly some transfer from POM to MAOM, much of MAOM formation occurs from DOM early during decomposition. There might just be a great abundance of DOM and labile inputs int La Cage to explain greater MAOM formation.*

Answer L483: We agree with this alternative explanation. We have therefore included it in the revised MS version.

See L514-518: « Cotrufo et al. (2015), shows that POM and MAOM likely form under different biochemical and physical pathways. While there is certainly some transfer from POM to MAOM, much of MAOM formation occurs from dissolve organic matter (DOM) early during decomposition. There might just be a great abundance of DOM and labile inputs in La Cage to explain greater MAOM formation."

*L486: 'Can mess up' could be replaced with a more formal term.*

Answer L486: We replaced this term with "hamper".

See L520: "POM heterogeneity can hamper SOC stability assessments"

*L509: However, a similar quantity of carbon is observed in the MAOM fraction as in La Cage correct? Could it simply be a result of higher structural inputs from OWP treatments?*

Answer L509: We believe that the differences in the distribution of additional carbon in the physical fractions would probably be associated with the quality and nature of the carbon inputs in both experiments. The application of PROs would potentially contribute more structural carbon than crop residue.

*L525 Is it surprising that carbon stored by OWPs is less stable than in baseline? Given that the baseline condition primarily retains the most stable and recently added carbon fractions, any newly added or altered carbon would inherently exhibit reduced stability.*

Answer L525: Thank you for your question. The "baseline" here is conventional control without organic inputs. In this system, the carbon inputs come from crop residues and this system has been in equilibrium since the beginning of the experiment as the site has been under cultivation for a long time. If the carbon in the conventional control were not more stable, we would have observed a drastic drop in carbon stocks. This was not the case.

*L540: Implementing thermal analysis techniques on physical fractions to ascertain the proportion of MAOM that is exchangeable or stable would be a valuable extension of this research. Such an approach could yield significant insights, particularly regarding the nitrogen provisioning capacity of MAOM in agricultural systems. This is in line with the findings presented by Jilling et al. 2021 in Soil Biology & Biochemistry, which highlight the potential implications of MAOM dynamics for nutrient management in agricultural contexts.*

Answer L540: Thank you for this important point. Thermal analysis of the fractions may provide valuable information. It should at least be pointed out that the partysoc machine learning model has only been trained on bulk soil in cropland. Extrapolating the model to fractions requires further development.

*The manuscript makes a substantial contribution to the field of soil science. The recommended revisions are aimed at enhancing its clarity, consistency, and the interpretive depth of its findings.*

Thank you for your thoughtful evaluation of our manuscript. We are pleased to hear that our work is considered to be a significant contribution to the field of soil science. We appreciate the constructive feedback provided. We have carefully considered your recommendations to improve the clarity, consistency and depth of interpretation of our results. These revisions will undoubtedly strengthen the overall quality and impact of our research.

Answer reviewer #2

The paper by Kpemoua et al. employs three methods to investigate the biogeochemical stability of soil organic carbon in long-term agricultural plots undergoing different management treatments at two experimental sites in France. The authors combine physical fractionation, thermal analysis, and soil incubation to assess the relative stability and turnover of fractions of the soil with varying mean residence times. Their results are interesting – they find that the mineralization rate of added carbon is dependent on treatment, and that the distribution of the soil organic carbon among physical fractions potentially reflects differences in the quality of inputs into the soil system (i.e., partially decomposed organic wastes contributed more to the particulate organic matter fraction than fresh inputs). The thermal analysis paired with machine learning indicated that the majority of carbon that was potentially added by the treatments was in the active pool, with a lower activation energy relative to compounds found in the stable pool. I really like the triangulation approach to understanding organic carbon dynamics in these systems, and I believe the authors have conducted an interesting and useful experiment here. However, I have a few hesitations around the interpretation of the data that I would like to see addressed prior to this MS moving forward. I first detail my major concerns below, and then provide specific line comments for the authors consideration.

We thank the reviewer for the positive comments. We addressed the issues raised by modifying some parts of the manuscript. Please find the details below.

My primary concern is around the repeated use of the term "additional carbon," and the assumptions made using this identified pool. From my understanding of the sampling approach, the authors took samples at only one time-point, such that the conventional treatments (CON-LC at La Cage, and CON-QA at QualiAgro) were used as proxies for the state of the soil either a.) if no intervention was ever put in place, or b.) at the start of the trial. I think in principle, this makes sense. However, from the text, I think it's unclear how the reader should interpret "additional" carbon. If it falls under scenario a., then the implication is that additional carbon should be interpreted as any carbon that the treatment plots have beyond what is present in the conventional plots, regardless of origin. If that's the case, then I think this just needs to be really clearly defined, and there just needs to be some discussion around the C contents of the soils across the treatments at the inception of the experiment, such that the reader understands that a fair comparison is being made. However, if the intention is to use the conventional treatment as a proxy for soils at the start of the experiment (scenario b.), then the implication is that any additional carbon is derived from the treatment (an interpretation substantiated in the text, see L79 and L125-126), and further information is needed still to verify that the conventional treatments were at steady state to ensure that the estimates of additional carbon are accurate. This ambiguity also stems in part from the use of "baseline" in describing the conventional treatments (see L206). I'd really appreciate the authors clarifying this term throughout the manuscript, as well as their intended interpretation of the conventional plots as controls to be compared against.

Thank you for these points. In this study, we defined additional carbon as the difference in carbon stock between the agroecological practices and the conventional practice for the same measurement date. It is true that the additional carbon calculated with this method does not take into account only new carbon from organic waste products or crop residues. However, both study sites were under cultivation many years before the experiments started. If we look at the carbon stock monitoring in these two experiments on conventional practices, we can see that carbon stocks are almost at equilibrium, because there is no significant variation in carbon stock between years since 1998 (see Autret et al., 2016 for La Cage and Peltre et al., 2012).

Regardless of the steady-state assumptions, I still have some hesitations around the calculations that are made using the additional carbon values assuming that additional carbon is "SOC stored upon the implementation of C storing practices". For example, in Section 3.2 (Lines 351 and on), the authors detail the distribution of the additional carbon among physical fractions. In my reading, it was not clear from the text in this section or in the methods section how this is done, but my interpretation from

looking at the supplement is that the difference between either the POM or MAOM (g C kg soil$^{-1}$) in the treatment and the conventional plots was calculated, and then divided by the difference in the bulk soil C between the two treatments; is this accurate? Assuming that's the case, it becomes challenging for me to consider this wholly additional carbon, as there isn't a way to directly attribute it to the treatment inputs (i.e., an isotope tracer). How can this method account for inter-fraction transfer of legacy carbon between fractions? If the rate at which legacy POM decomposes and contributes to the MAOM fraction is modified by treatment (if POM is primed by the new inputs of N rich litter in the alfalfa treatments, as an example) then couldn't this calculation show additional carbon in the MAOM fraction based solely on transfer, without any real treatment effect? My central concern here is that the implementation of a new treatment changes the dynamics of the soil system, such that part of the carbon that is present in the amended plots may not be due to the treatment (i.e., additional carbon), but instead is carbon that was present at the start of the experiment that is just being lost more slowly or changed in form. It gets back to this question—is additional carbon just carbon in addition to that in the conventional plots, or is it carbon added to the plots via the treatment?

The additional carbon of each fraction or pool was calculated in the same way as the additional carbon stock seen above. For example, to calculate the additional carbon stock of the POM fraction in conservation agriculture, we made the difference between the carbon stock of the POM fraction in conservation agriculture and the POM fraction in conventional agriculture (stockPOM$_{CA}$- stockPOM$_{CON-LA}$). So it's not the same calculation as you thought. We are fully aware that the additional carbon in our situation takes into account both new carbon and legacy carbon from biological modification. Unfortunately, as you well pointed out, the absence of isotopic tracers makes it impossible to distinguish the new carbon from the legacy carbon. Furthermore, if we assume that the conventional controls in these two experiments are in equilibrium, then a large part of the additional carbon would come from the effect of the practices, i.e. the increase of carbon inputs.

Finally, I think this issue of how additional carbon is handled is also problematic in the discussion of the mineralization data. From the methods in L317, "We assume that the extra carbon mineralized in the agroecological practice relative to the baseline practice comes from the additional carbon." I think this is a really important and not totally substantiated assumption—what of priming? Based on the experimental approach, there is no way to say for sure that the carbon respired in these incubations can be attributed to "new" vs. "old" carbon. The legacy carbon may make up 100% of what's being mineralized in the treatment plots-- to my understanding it doesn't really seem like there is way to definitively say one way or another. I think this ambiguity challenges the interpretation of the data somewhat and is something that needs to be addressed in the manuscript.

Thank you for your attention. It is indeed a bold assumption. If we take into account the analogy made to calculate additional carbon with other methods, it is nevertheless acceptable. We are aware that in this carbon mineralization pool, there is some new carbon and some legacy carbon induced by the priming effect. At the same time, for the other methods, additional carbon is a mixture of new and legacy carbon. This is the limitation of the incubation method. The absence of an isotopic tracer made it impossible to separate the new from the legacy carbon.

As I said above, I like the objectives and approach of this paper. I think it's a wonderful synthesis of methods that provides really interesting insight and nuance to the way we discuss fractions and what it means for carbon to be stable. But this calculation of additional carbon and the ambiguity of that term is really giving me pause, and I would like to see it addressed prior to publication.

**Line Edits**

L17: I found this sentence to be kind of confusing—which soils are, "these soils?" I'd suggest rephrasing for clarity.

Answer L17: Thank you for this point. We have rephrased this sentence as follows: "The implementation of agroecological practices often leads to additional soil organic carbon storage, and we have sought to assess the biogeochemical stability of this additional carbon."

L19: Particle, not particles.

Answer L19: We correct it. Thank you!

L34-36: If you want to say this, I think there needs to be a robust definition of "additional" in the abstract as discussed above.

Answer L34-36: Thank you for your suggestion. We have taken it into account in the revised version of the manuscript.

See L25-28: "The additional carbon resulting from agroecological practices is the difference between the carbon stock of the bulk soil, physical fractions or carbon pools in a soil under the agroecological practices and that of the same soil under a conventional practice as control."

L62: There is an unmatched parenthesis in this sentence, starting at … (crop residues, litter…

Answer L62: We correct it, thanks!

L75: This is where I think the ambiguity around the term additional could be addressed. From this section (L75 – L79), the implication to the reader seems like additional should be interpreted to mean "new" carbon.

Answer L75: We have clarified the term additional carbon through a definition and by nuancing that this additional carbon can contain both new and legacy carbon. You can see the modification in the revised version of the manuscript.

L77-81: "The additional carbon storage linked to agricultural practice B is the difference between the carbon stock in a soil under practice B and that of the same soil under a reference practice (Pellerin et al., 2019). This additional carbon storage is not necessarily the result of recent carbon inputs, but can also include the legacy carbon."

L88: Both POM and MAOM need to be initialized in the main text.

Answer L88: We have defined the acronyms POM and MAOM as underlined. Thanks for the suggestion.

See L93-95: "In a study comparing several fractionation methods, Poeplau et al. (2018) found that particle size fractionation was well suited to isolate particulate organic matter (POM) fractions from mineral associated organic matter (MAOM) with contrasting MRT."

L103: I agree with the other referee, I think it would be helpful to define kinetic carbon pools. Also, please be consistent about using abbreviations (i.e., C).

Answer L103: We decided to maintain the term "kinetic carbon pool" as it is an appropriate one. We define it as being a subset of the carbon pool to which we associate a residence time and where we consider that this subset has the same mean residence time.

L121-123: Same comment as above about C.

Answer L121-123: We have also answered this request above.

L139: Were these experiments converted from pasture, ley, etc. to agriculture at the start of the experiment?

Answer L139: Thank you for your question. The soils in these two experiments were under cultivation many years before the start of the experiments in 1998. So, there has been no conversion of grassland or pasture to agriculture.

L142 – 174: It would be helpful to the interpretation of the study and the results if these two subsections mirrored each other in the information that they provide. For instance, potential ET is provided for the La Cage site but not for the QualiAgro site. Information about the length of cultivation is only present for the QualiAgro site. Tillage information is presented only for one treatment, and it's unclear to me what the conventional system in CON-QA is from the text. I'd recommend the authors revisit this section and provide complete and equivalent site and treatment descriptions, if possible.

Answer L142-174: We have standardized the description of the sites to ensure the same information. You can therefore see the changes in the revised version of the manuscript of L149-L188.

L167: There's a typo here, three to tree.

Answer L167: Thanks for the remark. In the revised version of the manuscript, we have changed this sentence so that it no longer refers to three organic treatments. See L175: "Four treatments are considered in this study".

L170: It would be very helpful to understand the character of these various amendments, could please you provide the C:N ratios?

Answer L170: Thank you for your suggestion. Yes, it is indeed possible to have the C/N ratio of these organic amendments based to study of Obriot, (2016). We have included it as supplementary data in the manuscript (Table S5).

| Organic amendments | Corg | Norg | C/N | ISMO |
|---|---|---|---|---|
| | $(g.kg^{-1}MS)$ | $(g.kg^{-1}MS)$ | | (% Corg) |
| MSW | $310 \pm 45$ | $17.0 \pm 2.0$ | $18.5 \pm 4.1$ | $49.0 \pm 13.0$ |
| FYM | $324 \pm 67$ | $21.0 \pm 3.0$ | $15.5 \pm 2.7$ | $66.6 \pm 7.0$ |
| BIOW | $211 \pm 46$ | $16.9 \pm 4.1$ | $12.6 \pm 1.6$ | $75.5 \pm 6.3$ |

L187: I'm not sure I totally understand how the ESM calculations were done between the two sites. Were they done in a different manner, using samples from the start of the experimental period as the reference cores for the QualiAgro site, and contemporary samples from the CON-LA site as reference for the La Cage site? It's great that you were able to use ESM here and I think it makes the data more robust, but this section needs more information about where the data used to calculate ESM C stocks for La Cage were obtained from for the sake of repeatability.

Answer L187: Thank you for your question. In both experiments, we calculated carbon stocks based on equivalent soil mass (ESM) following the methodology outlined in Ellert and Bettany (1995). Therefore, we have revised the paragraph on stock calculation at La Cage to reflect this method as follows:

L211-215 "At La Cage experiment, the soil sampling strategy was designed to calculate SOC stocks on an equivalent soil mass (ESM) basis Ellert and Bettany (1995) over a depth at least equal to the deepest tillage event. The ploughing depth was *ca*. 30 cm before 1998 and shallower afterwards, about 25 cm (Autret et al., 2016). The sample was taken at the depth equivalent to a soil mass of 4300 kg. ha$^{-1}$. The carbon stocks were calculated by multiplying the SOC contents with this equivalent soil mass."

L206: I think the use of "baseline" here is misleading, and implies a temporal sampling scheme with a pre-treatment, baseline measurement. I'd recommend considering a change in terminology to something akin to "control".

Answer L206: Thank you for bringing this up. We don't have a time series in this study, because the additional carbon was calculated considering conventional agriculture as the control. Therefore, we have changed "baseline" to "conventional control" in the revised manuscript to avoid confusion.

L240: Were the samples ground prior to elemental analysis? Also, can you provide a rationale for freeze-drying the fine silt and clay fractions rather than drying them in the oven?

Answer L240: We chose to freeze-dry the fine silt and clay fractions for two reasons. Firstly, we obtained significant suspension quantities and wanted to avoid spending more time to drying them in the oven. Secondly, organic matter in these fractions can change at temperatures of 50°C in long-time drying in oven.

L317: As I detail above, I do not believe you can make this claim without a tracer of some sort. Thus, I think this language needs to be adjusted to represent the implications of the measurement that are within the scope of the experimental design.

AnswerL317: Thank you for this point of attention. Indeed, it's a bold assumption. If we take into account the analogy made to calculate additional carbon with the other methods, it's still comfortable. We are aware that in this mineralized carbon pool we have new carbon and legacy carbon induced by the priming effect. At the same time, for the other methods, additional carbon is a mixture of new and legacy carbon.

L321: To which data was the log transformation applied to? Please detail which variables were transformed for analysis where possible.

Answer L321: The transformed data concern only the cSand and fSand fractions. We have specified this in the revised version of the MS.

See L338-340 : "All data were tested for normality and homogeneity of variance. Log-transformation was applied to the data for the cSand and fSand fractions, as the transformation improved the normality and variance substantially"

L351: Here again, please see my comments above.

Answer L351: Thank you for your comment. We have re-phrased this sentence to indicate that the additional carbon is the result of a difference between the carbon stock of agroecological practices and the carbon stock of conventional control.

See L371-374: "We calculated the distribution of additional carbon (ΔSOC) in the fractions by difference of the carbon stock of the bulk soil or physical fractions under agroecological practices with the carbon stock of the bulk soil or physical fractions under conventional control (CON-QA and CON-LC respectively for the QualiAgro and the La Cage experiments)"

L368: Can you split this figure up in a way that allows for an interpretation of the smaller bars? Right now, it is really difficult to ascertain anything about SOC values for all but a few of the fractions. Further, there is an inconsistency in the way non-significant within fraction differences are displayed—on the top it seems like there non-significant differences are all grouped under a single letter, while on the bottom, NS is used to designate this. Finally, please add to the caption to specify the correspondence between the letters used in the plots and the data described in the caption, i.e., SOC stocks (Top, a., b.) … etc.

Answer L368: Thank you for your suggestion. Concerning the graphic, it would be difficult to split it again. It will be too heavy. We suggest putting the total stock and additional stock data in the supplementary data (See Table S3 and Table S4). Concerning the letters for differences, we have corrected it to avoid having "NS". See the revised version of the MS.

**Table. S3**: Carbon stock and additional carbon stock distribution in size and density fractions. Mean ± standard deviation of 4 replicates. Soils were sampled in all treatments with application of municipal solid waste (MSW) compost, biowaste compost (BIOW), farmyard manure (FYM), and the control without organic input (CON-QA).

| | Bulk soil | cPOM | fPOM | cSand | fSand | csilt | fSilt | Clay |
|---|---|---|---|---|---|---|---|---|
| **Carbon stock (t C. ha$^{-1}$)** | | | | | | | | |
| CON-QA | 39.31 ± 2.49 | 4.02 ± 1.73 | 4.04 ± 0.43 | 0.17 ± 0.04 | 0.24 ± 0.07 | 1.85 ± 0.23 | 0.84 ± 0.15 | 28.14 ± 2.80 |
| MSW | 54.03 ± 0.59 | 4.79 ± 0.91 | 7.55 ± 0.88 | 0.93 ± 0.65 | 0.17 ± 0.09 | 2.51 ± 0.17 | 1.09 ± 0.48 | 36.98 ± 1.53 |
| FYM | 54.77 ± 1.39 | 4.33 ± 1.52 | 9.71 ± 1.02 | 0.58 + 0.10 | 0.19 ± 0.07 | 2.45 ± 0.19 | 1.14 ± 0.50 | 36.37 ± 0.52 |
| BIOW | 63.17 ± 2.56 | 6.15 ± 1.14 | 9.48 ± 0.98 | 1.77 ± 0.53 | 0.25 ± 0.03 | 3.44 ± 0.50 | 1.05 ± 0.16 | 41.02 ± 3.41 |
| **Additional Carbon stock (t C. ha$^{-1}$)** | | | | | | | | |
| MSW | 14.72 ± 1.28 | 0.77 ± 0.98 | 3.51 ± 0.49 | 0.76 ± 0.33 | -0.07 ± 0.06 | 0.65 ± 0.14 | 0.26 ± 0.25 | 8.84 ± 1.60 |
| FYM | 15.46 ± 1.43 | 0.31 ± 1.15 | 5.66 ± 0.56 | 0.42 ± 0.06 | -0.05 ± 0.05 | 0.59 ± 0.15 | 0.30 ± 0.26 | 8.23 ± 1.42 |
| BIOW | 23.86 ± 1.79 | 2.12 ± 1.04 | 5.44 ± 0.54 | 1.60 ± 0.26 | 0.01 ± 0.04 | 1.59 ± 0.28 | 0.21 ± 0.11 | 12.88 ± 2.21 |

**Table. S4**: Carbon stock and additional carbon stock distribution in size and density fractions. Mean ± standard deviation of 4 replicates. Soils were sampled in conservation agriculture (CA), organic agriculture (ORG) and conventional agriculture (CON-LC).

| | Bulk soil | cPOM | fPOM | cSand | fSand | csilt | fSilt | Clay |
|---|---|---|---|---|---|---|---|---|
| **Carbon stock (t C. ha$^{-1}$)** | | | | | | | | |
| CON-LC | 42.21 ± 2.07 | 3.01 ± 1.24 | 4.22 ± 0.36 | 0.34 ± 0.22 | 0.40 ± 0.11 | 1.96 ± 0.20 | 2.67 ± 0.14 | 27.03 ± 1.99 |
| ORG | 44.66 ± 1.80 | 3.23 ± 0.07 | 4.57 ± 0.15 | 0.29 ± 0.11 | 0.45 ± 0.08 | 2.11 ± 0.27 | 2.81 ± 1.02 | 28.70 ± 2.16 |
| CA | 57.17 ± 4.53 | 3.54 ± 1.06 | 5.90 ± 0.72 | 0.49 + 0.10 | 0.43 ± 0.19 | 2.47 ± 0.29 | 2.41 ± 1.66 | 39.08 ± 2.58 |
| **Additional carbon stock (t C. ha$^{-1}$)** | | | | | | | | |
| ORG | 2.44 ± 1.38 | 0.23 ± 0.62 | 0.35 ± 0.19 | -0.05 ± 0.12 | 0.05 ± 0.07 | 0.15 ± 0.17 | 0.14 ± 0.52 | 1.67 ± 1.63 |
| CA | 14.95 ± 2.49 | 0.54 ± 0.82 | 1.68 ± 0.40 | 0.16 ± 0.12 | 0.03 ± 0.11 | 0.51 ± 0.18 | -0.26 ± 0.83 | 12.05 ± 1.47 |

L400: How was this calculated? How can you separate the additional carbon from the control treatments within the Party$_{SOC}$ model? This needs to be detailed in the methods. Looking at the plot, it seems like maybe the C stock of the CON treatment was subtracted from the C stock of the active and stable pools?

Answer L400 : The PARTYSOC model provided the active carbon and stable carbon values for all agricultural practices. To determine the additional carbon allowed by agroecological practices, we calculated the difference between the active carbon or stable carbon obtained under agroecological

practices and the values for the same pools under conventional control. We explained in the introduction how the additional carbon was obtained in this study.

See L25-28: "The additional carbon resulting from agroecological practices is the difference between the carbon stock of the bulk soil, physical fractions or carbon pools in a soil under the agroecological practices and that of the same soil under a conventional practice."

L413 and L416 : Typos, these should refer to Figure 4b and 4a, respectively.

Answer L413 and L416: We correct it!

L420: In addition to my comments above, I am having a very hard time understanding the calculations here. If there was lower C mineralization in the treatment plots at QualiAgro, and per line 317, only "extra carbon mineralized from the agroecological practice relative to the baseline comes from the additional carbon," how can you have 4-5% of the additional carbon mineralized?

Answer L420: Thank you for your comment. As stated in the materials and methods section (See L331-337), we observed that absolute carbon mineralization in agroecological practices was higher than carbon mineralization in conventional controls. This is to be expected, since the more carbon there is in the soil, the more it is mineralized. We hypothesized that the extra absolute carbon mineralization in agroecological practices would largely come from additional carbon. In the following calculation, we related this extra absolute carbon to the proportion of additional carbon. This calculation is in line with the other methods, which is quite logical given that we are comparing biogeochemical stability using a multi-method approach. We also point out that this extra mineralized carbon in the incubations may come from new or legacy carbon.

L426: In my opinion, this figure is somewhat misleading, and should be revamped. These values can't be compared apples to apples like this, I think they need to be separated by methodology. The current figure implies that each of these bars represents a proportion of the total, when in fact there's a great deal of double counting here. If we were to add it all up, the percentage of total C for some of these soils would be > 200% in some cases. These data are really cool and they tell us something really interesting, but I think to convey that accurately the figure needs to be clear in its message, as well as its caption.

Answer L426 : Thank you for your suggestion. We agree with your point about confusion. We have therefore reworked the figures, classifying the carbon fractions or pools by method. You will see this in the revised version of the manuscript.

L445: Do you think this has anything to do with the difference in mineral sorption potential between the two sites (i.e., 25% sand vs. 7% sand)? Or could there be any inter-fraction transfer during the fractionation process (such as from POM into MAOM, if the POM has a density greater than 1 g cm$^{-3}$?).

Answer L445: Thank you for your point. Our results indicated that the additional carbon was largely in the clays. However, the percentage of clay in these experiments is similar. We think that the difference is related to the quality of the carbon inputs. In the fractionation method used, the cut-off was made using different sieve sizes. As a result, if there is a density problem, it will be mainly in the isolation of coarse or fine POM from the sands. For this reason, we have grouped these 4 fractions together to obtain the POM-C fraction.

L448: See my comment above re: L142. Is the implication here that there is no tillage in the organic or conservation treatments?

Answer L448: Thank you for your comment. Annual ploughing is carried out in organic agriculture, whereas no ploughing is done in conservation agriculture. We have specified this in the description of practices.

L455: This line about 12 years is a rather large assumption—how can you prove that the transfer from POM-C to MAOM-C didn't occur after 6 years, or 8 years, or given the subsequent lines about rhizodeposition, occurred at all? I'd suggest reframing this interpretation somewhat.

Answer 455: Thank you for your comment. We have revised this sentence. Indeed, the transfer can start at 6 or 8 years. We have therefore modified the sentence as follows:

L482-487: "Interestingly, an earlier analysis of SOC distribution at La Cage, after 5 years of differentiation, showed a significant increase of POM-C in the conservation agriculture system, while no change of POM-C in the organic system and no significant change of the MAOM-C (Balabane et al., 2005), suggesting either that it took more than 5 years for the additional POM-C to be broken down and biodegraded as MAOM-C, or that the introduction of alfalfa as the cover crop instead of fescue since 2008 (i.e., 12 years later) resulted in more direct rhizodeposits inputs to MAOM-C"

L462: I'm not sure I understand what fine-sized OM is in this context (unless it's synonymous with MAOM?). Wouldn't rhizodeposits act as inputs into the dissolved organic matter pool?

Answer L462: In this section the "fine-sized OM" refers to the MAOM. We have specified MAOM to avoid confusion. Thank you for this point.

See L490-492 : "The cover crops and legume rotation in conservation agriculture and the legume rotations in organic agriculture would likely have affected carbon input via the root system as dead roots (POM) and rhizodeposits (MAOM)."

L480: What do you mean by "small POM" here?

Answer L480 : To make things clearer, we've reviewed what "small POM" means in the revised version.

See L506-513 "Peltre (2010) observed that the short-term application (4 times) of the OWPs at QualiAgro increased the additional carbon only in the POM-C fraction, the MAOM-C fraction <50µm being unchanged. Paesch et al. (2016) later found that 7 successive applications of the OWPs led to additional carbon in occluded small POM (< 20µm) and in the fine silt + clay fraction (< 6.3 µm). After 11 applications of OWPs we observed an increase in fine POM (50-200µm), coarse silt (20-50µm) and MAOM-C (<50 µm). This series of results indicate that the application of the OWPs increase in the short term the POM-C fraction and that in a longer term (> 10 years) the organic carbon in the POM-C is transferred to the MAOM-C through biological activity in the soil."

L486: I agree with the other referee, the title of this section should be changed.

Answer L486: We replaced the term "mess up" with "hamper".

See L520: "POM heterogeneity can hamper SOC stability assessments"

L504: In addition to the amendment C:N ratios, it could be helpful to see what the POM C:N ratios are for these treatments. If the amendment is driving a shift towards a more slowly decomposed POM fraction, does that show up in the stoichiometry? It's hard to guess at from the C:N ratios presented in Table 1, given the relative proportion of POM in these soils.

Answer L504: Thank you for your point. We looked at the C/N ratios of the cPOM and fPOM fractions, there was no striking difference between the agricultural practices at the two sites. This suggests that sttocheometry is not an explanatory factor. We have included the C/N data for the cPOM and fPOM fractions in the supplementary data.

[Figure]

L547: This is a really important and timely result, it could be nice to see it expanded on here and connected to the literature around a dynamic MAOM pool.

Answer L547:

In the Supplement, I think there's a typo with the POM, it's currently listed as MOP.

Answer: Thank you for this point. We correct it!